# Neuropelveology for Endometriosis Management: A Systematic Review and Multilevel Meta-Analysis

**DOI:** 10.3390/jcm13164676

**Published:** 2024-08-09

**Authors:** Leila Allahqoli, Sevil Hakimi, Zohre Momenimovahed, Afrooz Mazidimoradi, Fatemeh Rezaei, Seyedeh Zahra Aghamohammadi, Azam Rahmani, Ghazal Mansouri, Fatemeh Hadavandsiri, Hamid Salehiniya, Ibrahim Alkatout

**Affiliations:** 1Ministry of Health and Medical Education, Tehran 14357-13715, Iran; lallahqoli@gmail.com; 2Faculty of Health Sciences, Ege University, 35575 Izmir, Türkiye; sevil.hakimi@ege.edu.tr; 3Midwifery Department, Qom University of Medical Sciences, Qom 37136-49373, Iran; zmomeni@muq.ac.ir; 4Neyriz Public Health Clinic, Shiraz University of Medical Sciences, Shiraz 71348-14336, Iran; neycdu@sums.ac.ir; 5Research Center for Social Determinants of Health, Jahrom University of Medical Sciences, Jahrom 46199-74148, Iran; f.rezaei@jums.ac.ir; 6Department of Mathematics, Islamshahr Branch, Islamic Azad University, Islamshahr 67653-33147, Iran; zahraaghamohamadi@iau.ac.ir; 7Nursing and Midwifery Care Research Centre, School of Nursing and Midwifery, Tehran University of Medical Sciences, Tehran 14167-53955, Iran; arahmani@sina.tums.ac.ir; 8Department of Obstetrics and Gynecology, School of Medicine, Afzalipour Hospital, Kerman University of Medical Sciences, Kerman 76169-13555, Iran; mansori@kmu.ac.ir; 9Occupational Sleep Research Center, Baharloo Hospital, Tehran University of Medical Science, Tehran 14167-53955, Iran; fatemeh.hadavand@sbmu.ac.ir; 10Social Determinants of Health Research Center, Birjand University of Medical Sciences, Birjand 97178-53577, Iran; hamid.salehi@bums.ac.ir; 11Kiel School of Gynaecological Endoscopy, University Hospitals Schleswig-Holstein, Campus Kiel, Arnold-Heller-Str. 3, Haus 24, 24105 Kiel, Germany

**Keywords:** neuropelveology, endometriosis, pelvic pain, nerve

## Abstract

**Background**: Despite the availability of treatments such as surgery and hormonal therapy, women with endometriosis often endure chronic problems. This review aims to evaluate the effectiveness and safety of neuropelveology. **Methods**: In a systematic review with a meta-analysis, we searched three electronic databases: MEDLINE (PubMed), Scopus, Embase, and Web of Science (WOS). The search was conducted in January 2024 with no date or language restrictions using a carefully curated set of keywords. We conducted a comprehensive review, including all observational and clinical trials reporting data on neuropelveology approaches in the management of endometriosis, irrespective of geographical location. The studies included in our review were required to be published in peer-reviewed journals and be available in any language, with at least an abstract in English. The data of all included studies were summarized in excel (version 19) and were analyzed by Comprehensive Meta-analysis v3.3 (Biostat) and STATA (version 17). A multilevel meta-analysis was performed on studies with two arms (intervention and control) to evaluate the efficacy of neuropelveology in managing women with endometriosis. **Results**: After screening 476 records, 30 studies, published from 1952 to 2021, were included in this review, each employing various methodologies. The studies were divided into the following three categories: (a) efficacy of neurectomy or nerve resection (*n* = 20), (b) efficacy of neurolysis (nerve blocks) (*n* = 4), and (c) efficacy of neuromodulation (*n* = 6) in the management of endometriosis. Among the studies evaluating the efficacy of neurectomy or nerve resection, 10 studies (with 18 group comparisons) were included in the random-effects meta-analysis. Treatment success (not occurrence of pain) was higher with neurectomy vs. controls (RR = 0.497, 95% CI = 0.236 to 1.04, *p* = 0.06 (for experimental studies) and RR = 0.248, 95% CI = 0.14 to 0.43, *p* < 0.001 (for observational studies)), representing a 50% and 75.2% risk reduction in the recurrence of pain in experimental and observational studies, respectively. Similarly, neurolysis, particularly superior hypogastric plexus blocks and uterine nerve ethanol neurolysis, demonstrated encouraging outcomes in pain reduction and an improved quality of life for women with endometriosis. The efficacy of neuromodulation in managing endometriosis symptoms appears promising but requires further investigation. **Conclusions**: In conclusion, neuropelveology approaches, such as neurectomy, neurolysis, and neuromodulation, offer significant potential for pain reduction in endometriosis patients, albeit with risks of complications and high recurrence rates, necessitating careful patient selection and long-term monitoring.

## 1. Introduction

Endometriosis, characterized by the ectopic growth of endometrial tissues outside the uterine cavity, stands as a prevalent gynecological disorder affecting approximately 6–20% of reproductive-age women [1]. This estrogen-dependent condition carries significant morbidity and ranks among the primary causes of infertility, dyspareunia, and dysmenorrhea, with the latter being the most frequently reported complaint [2]. The associated pain can be divided into visceral pain caused by the autonomous nervous system and somatic pain caused by affection of the somatic nervous system. Most of the pain in the pelvis that is associated with endometriosis can be referred to the autonomous nervous system (sympathicus and parasympathicus) [3]. While pelvic pain predominates, women with symptomatic endometriosis may also experience pain in the lower back and lower extremities unrelated to the autonomous nervous system [4,5,6]. Notably, affected individuals face a heightened risk of enduring chronic pelvic, genital, and low lumbar pain, often accompanied by distal radiation pain due to sacral plexus or somatic nerve compression [7,8,9,10]. Despite available conservative treatments such as pain medication and hormonal therapy, along with central pain management, satisfactory outcomes remain elusive. Conversely, surgical interventions carry considerable risks of severe and potentially irreversible complications [11,12,13,14,15]. The emergence of neuropelveology as a distinct discipline dedicated to pelvic nervous system pathologies and enhanced neurologic diagnoses in chronic pelvic pain presents a promising avenue for intervention [10]. Encompassing various medical treatments and pelvic nerve surgeries, neuropelveology spans techniques from decompression and neurolysis to nerve reconstruction and resection (e.g., sciatic nerve endometriosis) to pelvic neurofunctional surgery [16]. In response to the growing demand for innovative therapeutic approaches and heightened medical community interest, the establishment of the International Society of Neuropelveology in 2014 aims to drive advancements in research, standardize diagnostic and therapeutic procedures, and facilitate ongoing medical education in neuropelveology (see www.theison.org) [10]. In light of this context, this study endeavors to review and analyze the role of neuropelveology in managing endometriosis. By assessing available data, it seeks to elucidate the efficacy and safety of neuropelveology.

## 2. Materials and Methods

This review adheres to the Preferred Reporting Items for Systematic Review and Meta-Analyses (PRISMA) guidelines [17] and was prospectively registered at PROSPERO with the registration ID CRD42024512390. The primary objective of this review is to evaluate the efficacy and safety of neuropelveology in addressing complications associated with endometriosis.

The PICO question guiding this analysis is structured as follows:

P—population: women diagnosed with endometriosis (as per the International Classification of Diseases (ICD)-11 code GA10, confirmed histologically); I—intervention: neuropelveology interventions including (a) neurectomy or pelvic nerve resection, (b) neurolysis (nerve block), and (c) an implantation of leads for neuromodulation; C—control: conservative surgery or non-intervention; O—outcome: effectiveness and safety of the interventions; S—study designs: clinical trials and observational studies.

### 2.1. Search Strategy and Information Sources

We conducted a thorough search for relevant articles in three reputable databases: PubMed/MEDLINE, Scopus, and Web of Science. The search was conducted in January 2024 with no date or language restrictions, using a carefully curated set of keywords. These keywords included “neuropelveology”, “neurosurgical techniques”, “endometriosis”, “intensity of pain”, “sexual function”, “health-related quality of life”, “complications”, and “adverse effects”. To optimize the search results, we employed MeSH keywords and Boolean operators (such as AND, OR). The reference lists of relevant papers were reviewed manually by one reviewer (L.A.). For transparency and reproducibility, the search strategy and a comprehensive list of search terms can be found in Appendix A.

### 2.2. Study Selection

The study selection process was facilitated using the EndNote software (version X9, Thomson Reuters), which aided in listing and screening studies. The selection process was divided into three distinct phases: screening, selection, and data abstraction. During the screening phase, two trained authors (L.A. and A.M.M.) independently evaluated titles and abstracts of the identified studies. A total of 221 articles were deemed potentially relevant and subsequently progressed to the full-text review stage. In the selection phase, two authors (L.A. and S.H.) independently assessed the full-text articles against the inclusion criteria using a checklist-style form. Articles meeting the inclusion criteria were included in the final analysis. To ensure consistency and resolve any discrepancies, a third expert author (I.A.) reviewed the full-text articles and addressed any inconsistencies or disagreements that arose during the selection process. This rigorous approach to study selection helped to minimize biases and ensured that only high-quality and relevant studies were included in the systematic review. We conducted a comprehensive review, including all observational and clinical trials reporting data on neuropelveology approaches in the management of endometriosis, irrespective of geographical location. Reference lists of included papers and relevant meta-analyses were manually searched. Additionally, we thoroughly assessed reports from abstracts and presentations at major gynecological meetings to mitigate the risk of a publication bias. It is worth noting that the studies included in our review were required to be published in peer-reviewed journals and be available in any language, with at least an abstract in English. Notably, we imposed no restrictions on the publication date, thereby ensuring a comprehensive consideration of relevant literature spanning diverse temporal contexts. This approach aimed to provide a robust and holistic analysis of the role of neuropelveology in addressing the challenges posed by endometriosis. Studies of an experimental nature, investigations utilizing human cadavers, letters to the editor, review articles, case reports, video articles, and those primarily focused on teaching techniques or introducing novel procedures were all excluded from our analysis. Additionally, studies lacking full-text availability and those addressing adenomyosis, myoma, or other gynecological pathologies were omitted.

Furthermore, studies involving combined medical interventions, such as gonadotropin hormone-releasing hormone analogs, were not considered. Participants with coexisting psychiatric illnesses, including depression, were also excluded from our study cohort. To elucidate our selection process, we employed a PRISMA flowchart (Figure 1), providing a visual representation of how studies were screened and included in our review.

### 2.3. Outcomes

The primary endpoint of this systematic review centered on treatment success, specifically defined as the proportion of women experiencing a resolution of pain symptoms (including dysmenorrhea, dyspareunia, pelvic pain, and dyschezia) following neuropelveology interventions. Treatment success was determined by the absence or non-recurrence of moderate-to-severe pain during the designated follow-up period.

Additionally, secondary endpoints encompassed various symptom changes post-neuropelveology procedures, such as alterations in pregnancy rates, sexual function, quality of life, subjective satisfaction reported by women, levels of anxiety, non-opioid analgesic consumption during menses, voiding symptoms at different postoperative intervals, as well as peri- and postoperative complications.

### 2.4. Data Synthesis and Extraction

The two reviewers independently extracted data from the included studies using a customized data extraction table created in Microsoft Excel. The data extracted from the included articles for further analysis encompassed various key elements, structured to facilitate a comprehensive understanding and comparison. These included the following:

Demographic information (title of the study, last author’s name, date of publication, and country of publication); characteristics of the sample (participants, sample size, and age distribution); study-specific parameters (details of the intervention (neuropelveology approach); details of the control group (conservative surgery); study design; agent (dosage and amount of agent for injection); outcomes measured and instruments used for assessment; surgery techniques applied (open, laparoscopic, robot-assisted laparoscopic); and duration of follow-up); and main results (categorization of results based on the type of intervention applied) and complications (intraoperative and postoperative complications). To organize and present this information effectively, tables were utilized to describe both the characteristics of the studies and the extracted data. These tables served as valuable tools for summarizing and synthesizing the findings, enabling a clear interpretation and comparison across studies. In instances where discrepancies arose between the two reviewers, a process of discussion and debate ensued until a consensus was reached. This collaborative approach ensured that any differences in data extraction were resolved, and a unified interpretation of the findings was achieved.

### 2.5. Assessment of Risk of Bias in Individual Studies

Depending on the type of investigation, four tools were used to assess the methodological quality of the studies:(a)The Cochrane Risk of Bias Tool (RoB 2) for randomized controlled trials does not have specific predefined cutoffs but rather assesses the risk of bias in various domains of a study. For each domain, the RoB provides options to assign a judgment of “low risk”, “high risk”, or “unclear risk” of bias [18].(b)The methodological index for non-randomized studies (MINORS) tool includes twelve questions and divides articles into two qualitative ranks (low and high risk of bias) [19].(c)The Newcastle–Ottawa Scale (NOS) for cohort studies was used. The quality of each study was rated using the following scoring algorithms: ≥7 points were considered as “good”, 2 to 6 points were considered as “fair”, and ≤1 point was considered as “poor” quality [20].(d)The National Institutes of Health’s (NIH) quality assessment tool for case series studies, last updated in July 2021, consists of nine questions and assigns three quality ratings (good, fair, and poor) [21].

Two independent reviewers (L.A., S.H.) scored each study, and any discrepancies between reviewers were resolved by a consensus that was reached after a discussion, with the possibility of consulting a third reviewer if necessary.

### 2.6. Data Analysis

All statistical analyses were conducted using Excel (version 19), Comprehensive Meta-analysis v3.3 (Biostat), and STATA (version 17). The meta-analysis was performed on studies with two arms (intervention and control) to evaluate the efficacy of neurectomy in managing women with endometriosis. The studies evaluated different types of pain, including dysmenorrhea, dyspareunia, pelvic pain, and dyschezia, often measured using a visual analog scale (VAS). In cases where studies reported multiple types of pain, each outcome was analyzed separately. To conduct the meta-analysis, women treated with neurectomy and conservative surgery were compared to those receiving conservative surgery alone (controls) using the risk ratio (RR) statistic. An RR > 1 indicated a higher risk of pain recurrence, while an RR < 1 indicated a lower risk of pain recurrence with neurectomy. The pooled estimate and 95% confidence interval (CI) were computed for each outcome, namely, dysmenorrhea, dyspareunia, pelvic pain, and dyschezia. A random-effects model was employed for the multilevel meta-analysis, acknowledging the distribution of true effects due to diverse study designs. The analysis encompassed three levels: (a) level 1, where effect sizes varied among individuals within the primary study (sampling error variance); (b) level 2, where effect sizes differed among various outcomes (dysmenorrhea, dyspareunia, pelvic pain, and dyschezia) within the primary studies; and (c) level 3, where effect sizes varied across different studies. Corresponding forest plots were constructed for experimental and observational studies to present individual study findings and pooled meta-analysis results, where applicable. The heterogeneity of outcomes was assessed using the I^2^ statistic, categorizing it as small (25%), moderate (50%), and large (75%) [22]. Significant heterogeneity was defined by a Cochran Q test, with *p* < 0.1 or I^2^ > 50% [23]. A sensitivity analysis was conducted. Publication bias was visually assessed with funnel plots and quantitatively assessed with Harbord’s test [24]. The prediction interval for true effect sizes was calculated.

## 3. Results

### 3.1. Search Results

A total of 476 publications, 45 of which were duplicate articles, were indexed in three or at least in two databases (Web of Science, PubMed, and Scopus). Following a review of titles and abstracts, 298 were excluded. Among the remaining articles, 164 were omitted due to their lack of alignment with the objectives of this study. We reached out to the authors of three studies for access to the full text or for a clarification of the reported results [25,26,27]. Furthermore, four studies were deemed eligible for inclusion in this review. However, due to a lack of data separation in the findings section, we were unable to include them in our analysis. [28,29,30,31]. Finally, the present review comprised 30 studies (Figure 1) utilizing various methodologies, including randomized or non-randomized controlled trials, prospective clinical case series, and retrospective studies. Some studies did not provide data about the study design. These studies were conducted in various countries, including the USA [32,33,34,35,36,37,38,39,40], Italy [26,41,42,43,44], Finland [45,46], Taiwan [47,48], Turkey [49,50], Australia [51], Iran [52], New Zealand [53], Germany [9], Poland [54], Switzerland [55], France [56], China [57], the UK [58], France, and Danmark [27]. The detailed characteristics of each study are summarized in Table 1, Table 2 and Table 3.

Among the reviewed studies in this systematic review, 10 studies (with 18 group comparisons) that evaluated the effectiveness of neurectomy on pain related to endometriosis were included in the meta-analysis [32,33,34,35,41,42,43,44,53,57].

### 3.2. Synthesis of Results

Based on the applied neuropelveology approach, studies were divided into the following three categories: (a) efficacy of neurectomy or nerve resection in the management of endometriosis (*n* = 20), (b) efficacy of neurolysis (nerve blocks) in the management of endometriosis (*n* = 4), and (c) efficacy of neuromodulation in the management of endometriosis (*n* = 6).

(a)Efficacy of neurectomy or nerve resection in the management of endometriosis:

In total, we found 20 studies, conducted between 1952 and 2017, that evaluated the effectiveness of presacral neurectomy (PSN) (*n* = 16) [32,33,34,35,36,37,38,41,42,43,44,47,48,51,54,57], uterine nerve ablation (UNA) (*n* = 2) [53,58], and a decompression or large resection of the sciatic nerve (>30% of the nerve) (*n* = 2) [27,55] in the management of endometriosis. Out of the 20, 10 studies were conducted in two groups (intervention vs. control), and the remaining 10 studies were conducted without control groups. One of the studies had two types of designs [35]. In various studies, women of different average age ranges were included, spanning from 27.8 to 32.5 years old. The duration of follow-ups for patients varied significantly across studies, ranging from 2 to 96 months. An overall view of the studies which considered neurectomy or nerve resection in the management of endometriosis is shown Table 1. The risk of bias was high for 6 studies, low for 2 studies, fair for 10 studies, and poor for 2 studies and there were some concerns for 1 study (Table 2).

Among 10 studies with 18 group comparisons (in some studies more than one outcome had been evaluated), the crude rates of treatment were 81% (374/462) with neurectomy and 61% (327/532) with controls at follow-ups. The heterogeneity in the treatment success rate among the studies was moderate (I^2^ = 46.9%, *p* = 0.113). In the random effects meta-analysis, the treatment success (not occurrence of pain) was higher with neurectomy vs. controls (RR = 0.497, 95% CI = 0.236 to 1.04, *p* = 0.06 (for experimental studies) and RR = 0.248, 95% CI = 0.14 to 0.43, *p* < 0.001 (for observational studies)), representing a 50% and 75.2% risk reduction in the recurrence of pain in experimental and observational studies, respectively. Based on the results of the experimental studies, this risk reduction was not statistically significant (Figure 2). The results of a one-study-removed analysis suggest that the meta-analysis conclusions were not significantly influenced by any single study (all *p* < 0.001).

The true effect size in 95% of all comparable populations falls in the interval of 0.04 to 6.06. The distribution of the true effects (prediction interval) is shown by Figure 3.

A funnel plot asymmetry was evident (Figure 4), and a quantitative assessment indicated a publication bias (*p* = 0.6081).

Sexual function was either unaffected or improved following neurectomy [37]. Additionally, a significant improvement in the quality of life was observed in the intervention group compared with controls [27,43,44]. Studies have indicated that neurectomy has no significant effect on pregnancy rates or infertility [33,34]. However, the probability of recurrence or symptom persistence following these procedures is a consideration. It was reported that the eight-year probability of recurrence was 81.8% (27 patients) in the laparoscopic presacral neurectomy group [48]. The results of the efficacy of neurectomy or nerve resection in the management of endometriosis are summarized in Table 2.

The safety of neurectomy or nerve resection in the management of endometriosis was evaluated in none of the studies in 1036 patients [27,32,37,41,42,44,48,51,55]. The peri- and post-complication rates were 7.33% (76 out of 1036 patients) with interventions vs. 0% with controls (*p* = 0.02). The most complications were related to constipation (41%, 31 out of 76 patients) and bladder impairment (31.6%, 24 out of 76 patients). Hypoesthesia, hyperesthesia, or allodynia was reported in 11.9% of the participants (9 out of 76 patients), and infections were seen in 6.6% of the participants (5 out of 76 patients). Bleeding, vaginal dryness, and re-admission for further surgery accounted for 2.6% of the participants (2 out of 76 patients) for each complication. One study only mentioned one case of failure in PSN without specifying the complication. The complications following neurectomy or nerve resection in the management of endometriosis are summarized in Table 2.

(b)Efficacy of neurolysis (nerve blocks) in the management of endometriosis:

In total, we identified four studies, conducted between 1995 and 2021, that evaluated the effectiveness of neurolysis (nerve blocks) in the management of endometriosis [39,49,50,52]. A total of 45 women with endometriosis underwent neurolysis across these studies, utilizing different agents. The studies included women of different average age ranges, spanning from 31.6 to 33.4 years. The duration of follow-up varied from 6 to 12 months across the studies. Three studies performed superior hypogastric plexus blocks [39,49,52], while Sönmez et al. conducted uterine nerve ethanol neurolysis in 2016 [50]. An overall view of the studies which considered neurolysis (nerve blocks) in the management of endometriosis is shown Table 3. The risk of bias was low for one study and fair for three studies (Table 4).

Neurolysis (nerve blocks) was able to reduce the pain score of patients (dysmenorrhoea, dyspareunia, and chronic pelvic pain) by between 4.41 and 7, while CS only reduced by 2.55 scores, and this difference was statistically significant. In Wechsler’s study, 80% (4/5) of the patients had considerable or complete pain relief [39]. Superior hypogastric plexus blocks resulted in a 33.3-point and 29.7-point increase in the sexual rating scale scores and EHP-5 score and a decrease of 6.3 points of analgesic consumption during menses [49]. The efficacy of neurolysis (nerve blocks) in the management of endometriosis is summarized in Table 4.

The safety of neurolysis (nerve blocks) in the management of endometriosis was evaluated in four studies. The peri- and post-complication rates were 33.3% (15 out of 45 patients) with neurolysis [39,49,52]. The most complications were related to constipation (73.3%, 11 out of 15 patients) and bruising of the injection (13.3%, 2 out of 15 patients). Abdominal pain and retention of urine each were 6.66% (1 out of 15 patients). The complications following neurectomy or nerve resection in the management of endometriosis are summarized in Table 4.

(c)Efficacy of neuromodulation in the management of endometriosis

In total, six studies, conducted between 2012 and 2023, evaluated the effectiveness of neuromodulation in managing endometriosis symptoms [9,26,40,45,46,56]. Neuromodulation was performed in a total of 74 women with endometriosis across the studies. Various methods were applied for neuromodulation, including sacral neuromodulation (SNM), a stimulation of the lumbar or sacral nerve roots by electrode implantation, and a respiratory-gated auricular vagal afferent nerve stimulation (RAVANS) versus a non-vagal auricular stimulation (NVAS). In studies employing sacral neuromodulation, the S3 nerve root (sometimes S4) was stimulated with a low electrical current via an electrode placed through the sacral foramen under local anesthesia. These electrodes were connected to an external stimulator [26,45,46]. In a study by Kolodziej et al., a lead was laparoscopically placed in direct contact with the affected nerve and with a permanent generator [9]. Nyangoh Timoh et al. placed an electrode unilaterally next to the S3 sacral nerve root and connected it to an external pacemaker, with the patients wearing the electrode and external neurostimulator for 21 days [56]. Napadow et al. conducted a study comparing RAVANS to NVAS, where the subjects were seated in a reclined position for both sessions, and modified press-tack electrodes were inserted in the left ear during the RAVANS stimulation session. The duration, intensity, pulse frequency, and other stimulation parameters were consistent between the RAVANS and NVAS sessions [40]. An overall view of the studies which considered neuromodulation in the management of endometriosis is shown Table 5. The risk of bias was some concern for one study and fair for five studies (Table 6).

The age range of women in the studies varied from 34.4 to 36.3 years, and the duration of patient follow-ups ranged from 15 min to 60 months. The results of the studies show that neuromodulation reduced the frequency and intensity of pain (pelvic pain, dysmenorrhea, and dyspareunia) and anxiety, and they show an improvement in symptoms and a better quality of life, and most patients expressed a desire to continue with SNM [9,40,45,46,56]. But in a study by Agnello et al., it was seen that 30.8% of the patients experienced no relief, leading to a removal of the system [26]. The characteristics of the studies that reviewed outcomes of neuromodulation in the management of endometriosis are summarized in Table 6.

The safety of neuromodulation in the management of endometriosis was evaluated in three studies [9,46,56], and the complication rate was 15% (11 out of 74 patients) with neuromodulation. The reported complications were infection (73%, 8 out of 11 patients), and pain was reported in 23% of the participants (3 out of 11 patients). The complications following neuromodulation in the management of endometriosis are summarized in Table 6.

## 4. Discussion

Pain often recurs after conservative surgery for endometriosis [59], a conclusion supported by this systematic review and meta-analysis. A synthesis of the results from the studies on neuropelveology approaches for managing endometriosis provides valuable insights into three main categories: neurectomy or nerve resection, neurolysis (nerve blocks), and neuromodulation. The efficacy of neurectomy or nerve resection, as evidenced by 20 studies spanning several decades, indicates a promising treatment option for reducing pain associated with endometriosis [27,32,33,34,35,36,37,38,41,42,43,44,47,48,51,53,54,55,57,58]. Despite varying surgical techniques and durations of follow-ups, the overall success rate of neurectomy compared to conservative surgery alone suggests a significant reduction in pain symptoms (representing a 50% and 75.2% risk reduction in recurrences of pain in experimental studies [35,41,43,44,53,58] and observational studies [32,33,34,35,42,57], respectively, with notable improvements in the quality of life and sexual function [27,43,44]). Despite a trend towards a reduction in pain recurrence with neurectomy compared to controls, the findings did not reach statistical significance. This suggests that while neurectomy may offer potential benefits in alleviating endometriosis-associated pain, the evidence from experimental studies alone is not strong enough to conclusively support its efficacy in this regard. However, it is essential to acknowledge the potential for complications associated with neurectomy, including but not limited to nerve injury, sensory disturbances, and the risk of pain recurrence [27,32,37,41,42,44,48,51,55]. Notably, the analysis revealed an eight-year recurrence rate of 81.8% [48], highlighting the need for ongoing monitoring and potentially adjunctive therapies to optimize long-term outcomes while mitigating the risk of complications. Furthermore, the high risk of bias in some studies and the potential for complications underscore the importance of careful patient selection and postoperative management.

Similarly, neurolysis, particularly superior hypogastric plexus blocks and uterine nerve ethanol neurolysis, demonstrates encouraging outcomes in pain reduction and an improved quality of life for women with endometriosis [39,52]. Though the number of studies is limited, the consistent findings across different agents and follow-up durations suggest the potential for neurolysis as an adjunctive therapy or an alternative for patients with refractory symptoms. However, it is important to note the occurrence of complications associated with neurolysis, such as constipation and bruising at the injection site, underscoring the need for careful patient selection and monitoring during these interventions [39,49,52].

In contrast, the efficacy of neuromodulation in managing endometriosis symptoms appears promising but requires further investigation. With only six studies conducted in recent years, employing various neuromodulation techniques, the evidence is still evolving [9,26,40,45,46,56]. However, initial results indicate potential benefits in pain reduction and functional improvement, particularly with sacral neuromodulation [9,40,45,46,56]. Further research with larger sample sizes and longer follow-up periods is warranted to establish the long-term efficacy and safety profile of neuromodulation in this patient population.

The variability in neuromodulation outcomes underscores the complexity of treating endometriosis-related pain and the potential influence of individual patient factors on treatment responses [46]. Moreover, the analysis highlighted the need for further research to elucidate the optimal parameters and patient selection criteria for neuromodulation approaches in this context.

Overall, while each neuropelveology approach offers unique advantages and considerations, the synthesis of available evidence underscores the importance of a multidisciplinary approach in managing endometriosis, tailoring treatment strategies to individual patient needs and prioritizing long-term symptom relief and quality-of-life outcomes. Further research and clinical trials are essential to elucidate the optimal role of neuropelveology interventions in the comprehensive management of endometriosis. Future studies should adopt comprehensive strategies for the design and reporting of randomized controlled trials, with a strong emphasis on standardization and methodological rigor [60]. Additionally, ensuring an adherence to standardized protocols and proper training in surgical techniques is essential for minimizing variability and improving the reliability of outcomes [61].

## 5. Limitations

This meta-analysis faces limitations related to the quality of the available studies, which may impact the interpretation of the results. The heterogeneity in the study designs and surgical techniques introduces potentially confounding factors that may affect the data analysis. Consequently, a generalization of the findings should be made with caution, acknowledging these limitations and the potential variability in the data.

Additionally, due to the limited number of studies and the inseparability of some information, we were unable to adequately assess factors such as surgical access (laparoscopy vs. laparotomy) and follow-up durations. To address these issues, future research should involve multicenter trials, where all participating centers adhere to a uniform surgical technique (e.g., laparoscopic neurectomy) and standardized follow-up schedules (e.g., assessments at 6 months, 1 year, and 5 years, post-surgery). This approach could help reduce variability and enable a more accurate comparison of outcomes between interventions and control groups. Future research efforts should focus on identifying predictors of treatment responses and developing personalized treatment algorithms to maximize the efficacy and safety of neuropelveology interventions in individuals with endometriosis-associated pain. Additionally, long-term randomized clinical trial studies are warranted to evaluate the durability of pain relief and the impact on the quality of life following neuropelveology procedures.

## 6. Conclusions

In conclusion, the systematic review and meta-analysis underscore the potential of neuropelveology approaches in alleviating endometriosis-related pain. Neurectomy or nerve resection, neurolysis, and neuromodulation offer promising avenues for symptom relief, with varying levels of evidence supporting their efficacy. While neurectomy demonstrates significant pain reduction, it comes with potential complications and high recurrence rates, highlighting the need for careful patient selection and long-term monitoring. Neurolysis presents as a valuable adjunctive therapy, albeit with associated complications that necessitate cautious management. Neuromodulation shows promise but requires further investigation to establish its long-term efficacy and safety profile. Overall, the findings emphasize the importance of a multidisciplinary approach in managing endometriosis, tailoring interventions to individual patient needs and prioritizing long-term symptom management and quality-of-life improvements. Further research, including larger clinical trials, is essential to elucidate the optimal role of neuropelveology interventions in the comprehensive management of endometriosis and to refine treatment strategies for better patient outcomes.

## Figures and Tables

**Figure 1 jcm-13-04676-f001:**
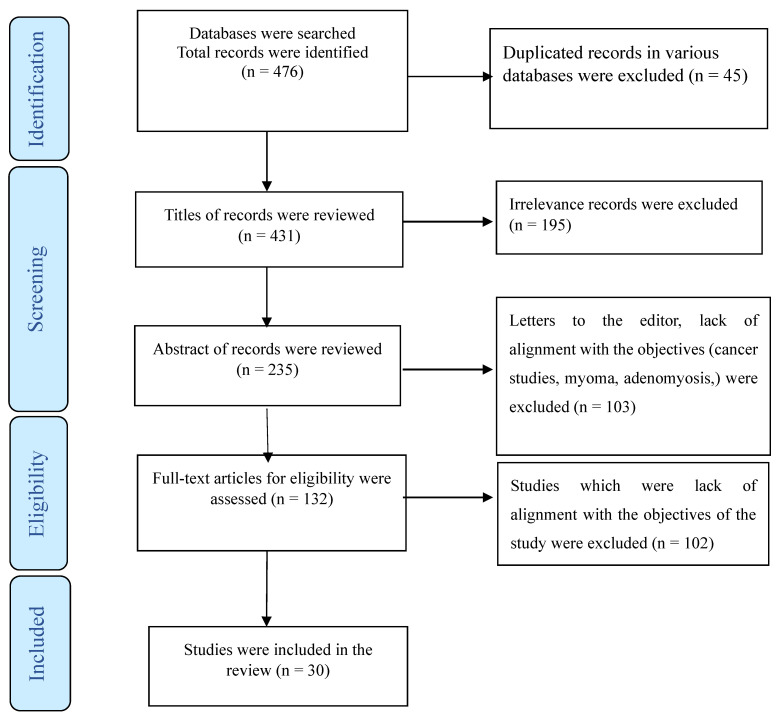
The process of screening and selecting relevant studies.

**Figure 2 jcm-13-04676-f002:**
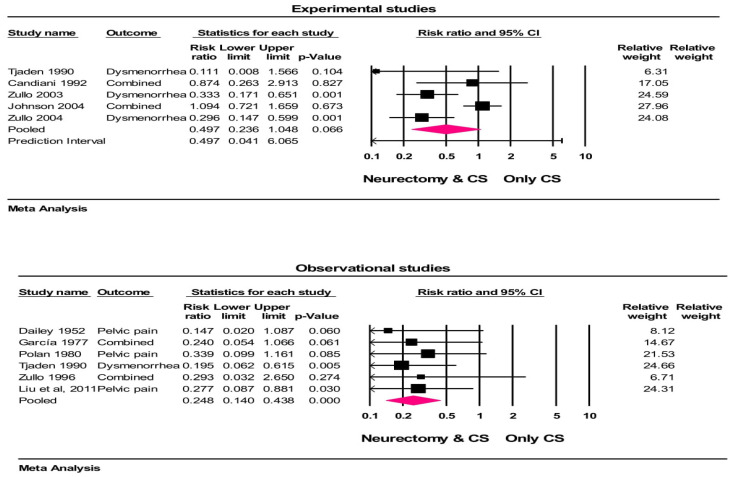
Forest plot of treatment success comparing conservative surgery with or without neurectomy. The risk ratio and 95% confidence interval are plotted for each study. The pooled risk ratio (diamond apex) and 95% confidence interval (diamond width) were calculated using a random effects model. A pooled risk ratio >1 suggests a higher risk with pain recurrence. A pooled risk ratio < 1 suggests a lower risk with pain recurrence. Random effects risk ratio: neurectomy vs. controls (RR = 0.497, 95% CI = 0.236 to 1.04, *p* = 0.06 (for experimental studies) and RR = 0.248, 95% CI = 0.14 to 0.43, *p* <0.001 (for observational studies)). Box size represents study weighting. Diamond represents overall effect size and 95% CIs. CS = conservative surgery [32,33,34,35,41,42,43,44,53,57].

**Figure 3 jcm-13-04676-f003:**
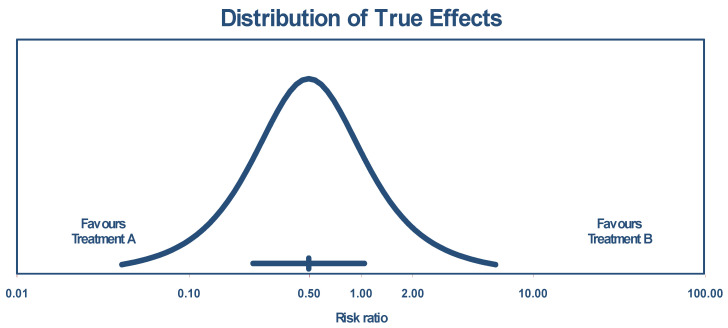
Distribution of true effects size based on experimental studies.

**Figure 4 jcm-13-04676-f004:**
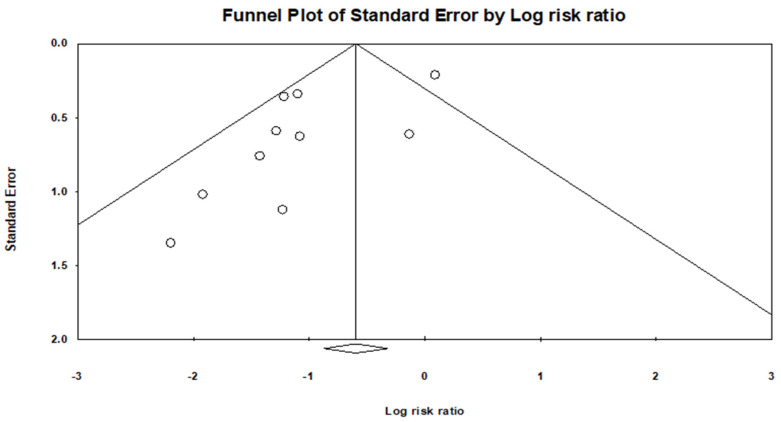
Funnel plot shows there is a publication bias.

**Table 1 jcm-13-04676-t001:** Overall view of studies which consider neurectomy or nerve resection in the management of endometriosis.

Characteristics		References
Study type	Observational studies	[27,32,33,34,35,36,37,38,42,47,48,51,54,55,57]
	Experimental studies	[35,41,43,44,53,58]
Producers	CS and PSN vs. only CS	[32,33,35,41,42,43,44,57]
	CS and UNA vs. only CS	[53,58]
	CS and PSN	[34,36,37,38,47,48,51,54]
	Large resection of the sciatic nerve (>30% of the nerve)	[27,55]
Follow-up duration	6 months	[35,42,58]
	12 months	[27,41,43,47,53,54]
	>12 months	[32,33,34,37,38,48,51,55,57]

CS: conservative surgery, PSN: presacral neurectomy, UNA: uterine nerve ablation.

**Table 2 jcm-13-04676-t002:** Characteristics of studies that reviewed outcomes of neurectomy or nerve resection in the management of endometriosis.

Study, Date, and Location	Participants	Sample Size (*n*)	Age	Neurosurgical Approach	Design	Technique	Variable	Questionnaire(s)	Duration of Follow-Up	Outcomes	Complications	Risk of Bias
Dailey et al., 1952, the USA [32]	Women with pelvic endometriosis	74 (CS + PSN, 30; only CS, 44)	18–33	CS + PSN vs. only CS	Retrospective	NA	Pain and fertility	NA	7 months to 60 months	CS + PSN vs. only CS: Complete pain relief: 80% vs. 45.5%. Partial pain relief: 16.7% vs. 31.8%. None: 3.3% vs. 22.7%. There was no effect of fertility.	Bladder impairment (*n* = 0) and failure in PSN (*n* = 1).	Fair
García and David 1977, the USA [33]	Women with endometriosis and infertility	71 (CS + PSN, 35; only CS, 36)	28.9	CS + PSN vs. only CS	Retrospective	Laparotomy	Dysmenourae, dysparunia, and pregnancy, occurance	NA	24 months	CS + PSN vs. only CS: Improved dysmenorrhea: 97.1% vs. 72.2%. Improved dyspareunia: 74.3% vs. 58.3%. Less impressive association of presacral neurectomy with pregnancy.	NA	Fair
Polan and DeCherney, 1980, the USA [34]	Women with endometriosis	CS + PSN (*n* = 8)	NA	CS + PSN	Retrospective	Laparotomy/laparascopy	Pain, pregnancy rate	NA	6 months to 84 months	75% of patients were pain-free. Pregnancy rate: 47%.	NA *	Fair
Tjaden et al., 1991a, the USA [35]	Women with endometriosis, stage III or IV	8 (CS + PSN, 4; only CS, 4)	31	CS + PSN vs. only CS	Randomized controlled study	NA	Dysmenourae and dysparonia	NA	6 months	CS + PSN vs. only CS: 100% vs. 0% had improved dysmenorrhea.	NA	High risk
Tjaden et al., 1991b, the USA [35]	Women with endometriosis, stage III or IV	18(CS + PSN, 13; only CS, 5)	31	CS + PSN vs. only CS	Non-randomized controlled study	NA	Dysmenourae and dysparonia	NA	6 months	CS + PSN vs. only CS: 85% vs. 0% had improved dysmenorrhea. There was a decrease in persistent back pain in the PSN group, but there was no improvement in lateral pain or dyspareunia.	NA	High risk
Candiani et al., 1992, Italy [41]	Moderate or severe endometriosis and midline dysmenorrhea	71 (CS + PSN, 35; only CS, 36)	32.5 (4.2) in the CS + PSN group; 31.1 (+3.6) in CS	CS + PSN vs. only CS	Randomized controlled study	Laparotomy or laparoscopy	Dysmenorrhea, pelvic pain, and deep dyspareunia intermenstrual pain	10-point VAS	12 months	CS + PSN vs. only CS: Moderate or severe dysmenorrhea was 80% vs. 75%. Deep dyspareunia and pelvic pain improved in 78.94% and 82.3% of patients compared with 77.8% and 80% of women who underwent CS.	Second laparotomy 2 days after presacral neurectomy for a retroperitoneal presacral hematoma (*n* = 1), constipation developed or worsened (*n* = 13), and urinary urgency (*n* = 3).	High risk
Nezhat and Nezhat, 1992, the USA [36]	Women with endometriosis	52	18–45 years	CS + PSN	Observational study	Laparoscopy	Pelvic pain and dysmenorrhoea	NA		Overall pain relief was reported by 49 (94%) of 52 patients. Dysmenorrhoea was reduced in 48 patients (92%).	NA	Fair
Perry and Perez, 1993, the USA [37]	Women with endometriosis	72	27.8 (6.1)	CS + PSN	Prospective consecutive cohort	Laparascopy	Pelvic pain sexual function	10-point VAS	24 months	Reduction in pain was reported in 91% of the patients. The preoperative pain score decreased from 7.9 ± 1.7 to 2.1 ± 1.4 at the end of 24 months. Sexual function was unaffected or improved.	Second laparotomy due to surgical complications (*n* = 1) for the control of bleeding. Vaginal dryness (*n* = 2), and constipation (*n* = 1).	Fair
Zullo et al., 1996, Italy [42]	Women with endometriosis	40 (CS + PSN, 16; CS + USR, 24)	29.3 (3.9) in CS + PSN vs. 31.2 (4.2)	CS + PSN vs. CS + USR	Retrospective study	Laparoscopy	Dysmenorrhea, deep dyspareunia, and pelvic pain	10-point VAS	6 months	CS + PSN could reduce the score of dysmenorrhea, dyspareunia, and pelvic pain to 4.5 ± 1.5, 4 ± 1, and 4.81 ± 1.88, respectively. CS + USR could reduce the score of dysmenorrhea, dyspareunia, and pelvic pain to 2.4 ± 1.59, 5.11 ± 2.23, and 4.7 ± 1.33, respectively. The efficacy of PSN in reducing pain was significantly higher than USR.	Major bleeding (>500 mL) from midsacral vessels, constipation (*n* = 5), and urinary urgency (*n* = 2).	Fair
Chen and Soong, 1997, Taiwan [47]	Women with endometriosis	335	30.9 (6.4)	CS + PSN	Retrospective study	Laparoscopy	Pain	5-point VAS	12 months	A total of 59% of the patients reported significant pain relief (no pain or mild pain requiring no medication).	NA *	Fair
Nezhat et al., 1998, the USA [38]	Women with pelvic pain attributed to endometriosis	176	Median age, 30 (18–45)	CS + PSN	A prospective postoperative follow-up	Laparascopy	Pelvic pain, dysmenorrhea, and dyspareunia	4-point scale (no, <50%, 50–80%, or >80% reduction)	12–72 months	Pelvic pain, dysmenorrhea, and dyspareunia were reportedly significantly reduced in 74%, 61%, and 55% of the patients, respectively.	NA	Fair
Kwok et al., 2001, Australia [51]	Women with endometriosis	9	28.2	CS + PSN	A retrospective case series	Laparascopy	Pain	4-point VAS	14.6 months (2–32 months)	A total of 77.8% of the patients reported significant improvement of symptoms. A total of 11.1% of the patients experienced mild improvement, while 11.1% failed to show any improvement in symptoms.	Mild loss of bladder filling sensation (*n* = 1) and constipation (*n* = 1), which lasted two weeks in one patient.	Poor
Sutton et al., 2001, the UK [58]	Women with pelvic pain and pelvic endometriosis	51 (laser vaporization + LUNA, 27; only laser vaporization, 24)	28 years (range, 20–41)	Laser vaporization + LUNA vs. laser vaporization	A prospective randomized double-blind controlled trial	Laparascopy	Dysmenorrhoea, dyspareunia, and chronic non-menstrual pelvic pain	10-point VAS	6 months	Significant differences in favour of the non-LUNA group for dysmenorrhea and non-menstrual pain. There were no significant differences recorded for dyspareunia.	NA	High risk
Zullo et al., 2003, Italy [43]	Women with dysmenorrhea due to endometriosis	126 (CS + PSN, 63; only CS, 63)	30.3 (3.7) in CS+ PSN vs. 31.9 (4.9) in CS only	CS + PSN vs. only CS	Randomized controlled trial	Laparoscopy	Dysmenorrhea, dyspareunia, chronic pelvic pain, and quality of life	100 mm VAS	12 months	Cure rate was significantly higher in CS + PSN group (85.6%) than in CS group (57%). Severity was significantly lower in CS + PSN group than CS group.	NA	High risk
Johnson, 2004, New Zealand [53]	Chronic pelvic pain in women with endometriosis	67 (CS + PSN, 32; only CS, 35)	NA	CS + LUNA vs. only CS	A prospective, double-blind randomized controlled trial	Laparascopy	Non-menstrual pelvic pain, dysmenorrhoea, deep dyspareunia, and dyschezia	10-point VAS	12 months	There was no significant difference in non-menstrual pelvic pain, deep dyspareunia, or dyschezia in women with no endometriosis undergoing LUNA versus no LUNA. The addition of LUNA to the laparoscopic surgical treatment of endometriosis was not associated with a significant difference in any pain outcomes.	NA	Low risk
Zullo et al., 2004, Italy [44]	Women with dysmenorrhea due to endometriosis	120 (CS + PSN, 60; only CS, 60)	31.9 ± 4.9 in CS+ PSN vs. 30.3 ± 3.7 in CS	CS + PSN vs. only CS	Randomized controlled trial	Laparascopy	Cure rates; frequency and severity of dysmenorrhea, dyspareunia, and chronic pelvic pain; and quality of life	Quality of life assessment (SF-36)	24 months	Cure rate was significantly higher in the CS + PSN group (83.3%) than in the CS group (53.3%). Severity was significantly lower in the CS + PSN group than the CS group. A significant improvement in quality of life was observed in the CS + PSN group compared with the CS group.	Constipation (*n* = 9), urinary urgency (*n* = 3), and both constipation and urinary urgency (*n* = 1).	Some concerns
Chang et al., 2007, Taiwan [48]	Women with dysmenorrhea due to endometriosis	88 (CS + LPSN, 33; CS + MLPSN, 55)	31.03 (5.9) in CS + LPSN vs. 31.16 (6.23) in CS+ MLPSN	CS+ LPSN vs. CS+ MLPSN	Non-randomized prospective study	Laparoscopy	Dysmenorrhea	5-point scale	96 months	Both groups demonstrated a significant decrease in the mean pain score when compared to the pre-surgery mean pain scores from 3.04 (1.04) to 1.04 (0.86) and from 2.91 (0.83) to 0.48 (0.46). Additionally, the eight-year probability of recurrence was higher in the LPSN group (81.8%, 27 patients) compared to the MLPSN group (43.6%, 24 patients).	The minimal peri-postoperative blood loss (*n* = 1), infections (*n* = 5), and constipation (*n* = 1).	Low risk
Jedrzejczak et al., 2009, Poland [54]	Women with endometriosis	16	30.3 (7.9); range, 21–46	CS + PSN	Cohort	Laparoscopy	Dysmenorrhoea and pelvic pain	VAS	12 months	At 12 months, dysmenorrhea increased. Pelvic pain not related to menses decreased by 67%. Dyspareunia declined dramatically at 3 and 12 months to a median score of 0.	NA	Poor
Liu et al., 2011, China [57]	Women with endometrisos	59 (CS + PSN, 28; only CS, 31	NA	CS + PSN vs. only CS	Non-randomized prospective study	Laparoscopy	Pain relief	NA	12.8 months (6 to 18 months	The post-operative pain relief rate was 89.28% (25/28) in the LPN group and 61.29% (19/31) in a control group.	NA	High risk
Possover, 2017, Switzerland [55]	Women with DIE	46	28 years (range, 24–36)	CS + a large resection of the sciatic nerve (>30% of the nerve)	Prospective clinical case series	Laparoscopy	Pelvic pain	10-point VAS	60 months	Pain was reduced from 9.33 preoperatively to an average 1.25 at a 3-year follow-up.	No perioperative or postoperative major occurred.	Fair
Roman et al., 2021, France and Denmark [27]	Women with DIE	52	NA	CS + decompression or a large resection of the sciatic nerve	Retrospective case series	Laparoscopy	Quality of life	Short-form 36 questionnaire	12 months	There was a significant improvement in the quality of life.	Bladder impairment (*n* = 14), hypoesthesia, hyperesthesia, or allodynia (*n* = 9).	Fair

CS: conservative surgery; DIE: deeply infiltrative endometriosis, LPN: laparoscopic neurectomy, LPSN: laparoscopic presacral neurectomy, MLPSN: modified laparoscopic presacral neurectomy, NA: not available, PSN: presacral neurectomy, VAS: visual analog scale, USR: ureterosacral resection, LUNA; laparoscopic uterine nerve ablation. * Complication was reported for all participants and could not be separated from other groups.

**Table 3 jcm-13-04676-t003:** Overall view of studies which considered neurolysis (nerve blocks) in the management of endometriosis.

Characteristics		Reference
Study type	Observational study	[39,49,50]
	Open-label pilot clinical trial	[52]
Producers	Superior hypogastric plexus block	[39,49,52]
	Uterine nerve ethanol neurolysis + CS vs. only CS	[50]
Agent	Diatnizoate meglumine + bupivacaine hydrochloride	[39]
	Phenol	[49]
	Ethanol	[50]
	Bupivacaine + triamcinolone	[52]
Follow-up duration	6 months	[50,52]
	12 months	[49]
	NA	[39]

CS: conservative surgery, NA: not available.

**Table 4 jcm-13-04676-t004:** Characteristics of studies that reviewed outcomes of neurolysis (nerve blocks) in the management of endometriosis.

Study, Date, and Country	Participants	Sample Size (*n*)	Age	Neurosurgical Approach	Design	Agent	Variable	Instrument	Duration of Follow-Up	Outcomes	Complications	Quality of Article
Wechsler et al., 1995, the USA [39]	Women with endometriosis and pelvic pain	5	28–31 years (mean, 31.6 years)	Superior hypogastric plexus block	Retrospective case series	A total of 1 mL of diatnizoate meglumine mixed with 6–8 mL of 0.25 bupivacaine hydrochloride	Pain	10-point VAS	NA	A total of 20% (1/5) of the patients experienced mild pain relief. A total of 60% (3/5) of the patients reported considerable pain relief. A total of 20% (1/5) of the patients achieved complete pain relief.	Abdominal pain (*n* = 1).	Fair
Soysal et al., 2003, Turkey [49]	Women with endometrisos	15	33.4 (1.9)	Superior hypogastric plexus block	Prospective observational study	A total of 10 mL of phenol	Pelvic symptom resolution, non-opioid analgesic consumption during menses, and sexual performance	3-point VAS and SSRS	12 months	A significant reduction was observed in the total pelvic symptom score, decreasing from 9.04 (1.2) pre-operatively to 2.2 (0.8) at the 12th postoperative month. SSRS increased from 30.9 (2.3) to 64.2 (2.8). Analgesic consumption during menses (ACDM) decreased from 8.9 (1.1) to 2.6 (0.8).	Retention of the urine (*n* = 1) and constipation (*n* = 11).	Fair
Sönmez et al., 2016, Turkey [50]	Women with pelvic pain	29 (9, endometriosis; 20, control group)	NA	Uterine nerve ethanol neurolysis + CS vs. only CS	Cohort	A total of 5 mL of 50% ethanol	Pain	10-point VAS	6 months	Preoperative VAS scores in the LUNEN group decreased from 7.59 ± 1.29 to 3.18 ± 2.88, postoperatively. In the control group, the VAS scores decreased from 7.90 ± 1.58 to 5.35 ± 3.09, postoperatively.	No problem.	Fair
Khodaverdi et al., 2021, Iran [52]	Women with endometriosis	16	33 ± 5.5, ranging from 26 to 39 years	Superior hypogastric plexus block	Open-label pilot clinical trial	A total of 10 mL of bupivacaine (0.25%), along with 40 mg of triamcinolone	Chronic pelvic pain, dysmenorrhoea, dyspareunia, and quality of life	10-point VAS and short form of EHP-5	6 months	The mean VAS score for dysmenorrhea improved significantly from 8.6 to 2.2. The mean VAS score for dyspareunia improved significantly from 7.8 to 1. The mean VAS score for chronic pelvic pain (CPP) improved significantly from 8.1 to 3. The mean overall baseline EHP-5 score was 54.3 ± 18.2 before the intervention and 24.6 ± 13.3 after the intervention.	Bruising of the injection site (*n* = 2).	Low risk

ACDM: analgesic consumption during menses, CS: conservative surgery, VAS: visual analog scale, SSRS: sexual rating scale.

**Table 5 jcm-13-04676-t005:** Overall view of studies which considered neuromodulation in the management of endometriosis.

Characteristics		References
Study type	Randomized, crossover, pilot study	[40,45,46]
	Observational study	[9,26,56]
Producers	RAVANS vs. NVAS	[40]
	SNM	[26,45,46,56]
	The lumbar or sacral nerve	[9]
Follow-up duration	15 min	[40]
	6–8 weeks	[26]
	12 months	[46]
	>12 months	[9,45,56]

RAVANS: respiratory-gated auricular vagal afferent nerve stimulation, SNM: sacral neuromodulation.

**Table 6 jcm-13-04676-t006:** Characteristics of studies that reviewed outcomes of neuromodulation in the management of endometriosis.

Study, Date, and Location	Participants	Sample Size (*n*)	Age	Neurosurgical Approach	Design	Variable	Instrument	Duration of Follow-Up	Outcomes	Complications	Quality of Article
Napadow et al., 2012, the USA [40]	Women with endometriosis	15	36.3 years old (10.6), range = 20–58 years)	RAVANS vs. NVAS	Randomized, crossover, pilot study	Deep tissue pain intensity, temporal summation of pain, and anxiety ratings	VAS—100 mm	15 min after stimulus cessation	RAVANS reduced the evoked pain intensity, temporal summation of mechanical pain, and anxiety compared to NVAS.	NA	Some concern
Nyangoh Timoh et al. 2015, France [56]	Women with endometriosis	5	34.4 years (SD: 4.7)	SNM	Retrospective study	Persistent voiding dysfunction	Electrophysiological and urodynamic tests	52 months	A total of 40% of the patients had a positive SNM test and underwent permanent implantation.	No complications	Fair
Lavonius et al., 2017, Finland [45]	Women with endometriosis	4	42.5	SNM	A pilot study	Symptom improvement and women’s subjective satisfaction	5- and 10-point VAS	30 months	A total of 75% of the patients reported a better quality of life and expressed a desire to continue with SNM.	NA	Fair
Kolodziej et al., 2019, Germany [9]	Women with endometriosis	6	Median age was 36.5 years	The lumbar or sacral nerve roots stimulated by an implantation of the electrode	Case series	Pain intensity, generic health status, and client satisfaction	10-point NRS, EQ-5D-5L PCS, and CSQ-8	7–60 months	The NRS improved from a median of 9.5, preoperatively, to 3.0. The median EQ-5D-5L index value before treatment was 0.18, indicating a notably lowered quality of life, and increased up to 0.83 after six months. Preoperative PCS was elevated with a median score of 41.0 and decreased to 4.0 after six months. CSQ showed that patients were satisfied with the treatment.	Infection (*n* = 1)	Fair
Agnello et al., 2021, Italy [26]	Women with endometriosis	13	NA	SNM	Retrospective study	Voiding symptoms	UDSs	6–8 weeks	A total of 69.2% of the patients experienced a significant improvement of symptoms that led to definitive implants. A total of 30.8% of the patients had no symptom relief, and the system was removed.	NA	Fair
Zegrea et al., 2023, Finland [46]	Women with endometriosis	31 patients	36 (19–59) years	Two stages of SNM	Multicenter prospective pilot	Postoperative pain and quality of life	BPI (0–10 points), CGI-I (0–7 points), 15D-measure of health-related quality of life (0–1 point), and B&B score (0–9 points)	12 months	Daily pain decreased from a median of 9 to 5. There was a statistically significant change in the overall 15D score. The median B&B score also improved significantly, decreasing from a baseline value of 8 to 4.5.	Stage 1: infection (*n* = 1) and pain (*n* = 2). Stage 2: infection (*n* = 5) and pain (*n* = 1).	Fair

BPI: brief pain inventory; B&B score: Biberoglu and Behrman, 1981; CGI-I: clinical global impression improvement; CSQ-8: client satisfaction questionnaire; EQ-5D-5L: the EuroQol 5-dimension 5-level instrument; NRS: numerical rating scale; NVAS: non-vagal auricular stimulation; PCS: pain catastrophizing scale; RAVANS: respiratory-gated auricular vagal afferent nerve stimulation; SNM: sacral neuromodulation; UDSs: urodynamic studies; VAS: visual analog scale.

## Data Availability

The data that support the findings of this study are available from the corresponding author upon reasonable request.

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
