# Peer review of "Neuropelveology for Endometriosis Management: A Systematic Review and Multilevel Meta-Analysis"

_jcm, 2024, doi:10.3390/jcm13164676_

Round 1

Reviewer 1 Report

Comments and Suggestions for Authors

The authors referred to a new and interesting field that is under development and will gain more importance in the future.
They did an overview and analysis of previous publications in a good way.
Apart from minor corrections, I have no major complaints

Line 39 and 112, change formulation, surgery is not conservative treatement modality.

Author Response

The authors referred to a new and interesting field that is under development and will gain more importance in the future.
They did an overview and analysis of previous publications in a good way.
Apart from minor corrections, I have no major complaints

Dear Reviewer,

Thank you for your time and efforts. We appreciate your valuable comments. The manuscript has been revised accordingly based on your feedback.

Line 39 and 112, change formulation, surgery is not conservative treatement modality.

Thank you for pointing out this comment. The revision was made based on this feedback.

Reviewer 2 Report

Comments and Suggestions for Authors

This  review aims to evaluate the effectiveness and safety of neuropelveology in treating endometriosis and includes all observational and clinical trials reporting these approaches. The initial identification of 476 publications is comprehensive.  The meta-analysis methodology is consistent, with clear reporting of treatment success rates and heterogeneity. Reporting relative risk (RR) with confidence intervals (CI) and p-values for different study types is appropriate. Detailed reporting of complications for different interventions is included. The conclusion underscores the potential of neuropelveology strongly.  Although, the review is very well written and descriptive, there are certain points which may help the authors to improve upon it.

1.      The removal of 45 duplicates is suitable but directs a need for better initial filtering to avoid duplicates.

2.      The omission of 298 articles after title and abstract review reflects a demanding initial screening process.

3.      The study uses various study designs (randomized/non-randomized trials, case series, retrospective studies) providing a broad perspective but this introduces heterogeneity.

4.      The review extents a wide temporal range (1952-2023), which could lead to inconsistency due to changes in medical practices over time.

5.      The authors have mentioned high risk of bias in six studies and fair/poor quality in others, however, the high complication rate with neurolysis and neuromodulation permits careful understanding.

6.      Although, neurectomy, neurolysis and neuromodulation shows promise, significant complications and high recurrence rates signifies that it needs further research to establish efficacy and safety.

7.      This study needs long-term randomized trials and larger sample sizes addressing the current study limitations with importance on personalized treatment algorithms and predictors of treatment response in patients.  

8.      Future studies identifying limitations such as study heterogeneity (study design), variability in surgical techniques (e.g., laparoscopy vs. laparotomy), and limited follow-up data is critical and necessary for contextualizing findings of this study. For instance, a multicenter trial could be designed where all participating centers follow a uniform surgical technique (e.g., laparoscopic neurectomy) and standardized follow-up schedules (e.g., assessments at 6 months, 1 year, and 5 years post-surgery). This approach may help reduce variability and allow for a more accurate comparison of outcomes between the intervention and control groups.

9.      Authors may consider the following papers to be included as references for suggesting future research directions and methodologies.

a.       Design and Implementation by Schulz KF, Altman DG, Moher D et a Lancet; 2001; 357:1191-1194. This paper provides complete strategies on the design and reporting of randomized controlled trials, emphasizing the importance of standardization and rigor in clinical research.

b.      The Importance of Protocol Adherence and Training" by Clavien PA, Barkun J, de Oliveira ML, et al Ann Surg. 2009;250:187– 196. This article discusses the tests and solutions in standardizing surgical techniques in clinical trials, which is crucial for reducing variability and improving the reliability of outcomes.

c.       The Long-term Impact of Surgical Interventions on Endometriosis-Related Pain and Quality of Life: A Systematic Review and Meta-Analysis" by Vercellini P, Viganò P, Somigliana E, et al. This systematic review and meta-analysis focus on the long-term outcomes of surgical interventions for endometriosis, providing valuable insights into the durability of pain relief and quality of life improvements.

Author Response

Reviewer#2

This  review aims to evaluate the effectiveness and safety of neuropelveology in treating endometriosis and includes all observational and clinical trials reporting these approaches. The initial identification of 476 publications is comprehensive.  The meta-analysis methodology is consistent, with clear reporting of treatment success rates and heterogeneity. Reporting relative risk (RR) with confidence intervals (CI) and p-values for different study types is appropriate. Detailed reporting of complications for different interventions is included. The conclusion underscores the potential of neuropelveology strongly.  Although, the review is very well written and descriptive, there are certain points which may help the authors to improve upon it.

Dear Reviewer,

Thank you for your thorough review and valuable feedback. We appreciate your positive remarks regarding our study.

We have taken note of your suggestions for improvement and have revised the manuscript accordingly.

Thank you once again for your time and efforts in reviewing our work.

The removal of 45 duplicates is suitable but directs a need for better initial filtering to avoid duplicates.

Thank you for your insightful comments. We appreciate your feedback on the need for better initial filtering to avoid duplicates. Unfortunately, there is no way to avoid this issue because, for a meta-analysis, it is necessary to search various scientific databases comprehensively to ensure that no relevant articles are missed. Some articles are indexed by three or at least two databases ((Web of Science, PubMed, and Scopus)

For clarification we have revised the sentence: “A total of 476 publications, 45 of which were duplicate articles, were indexed in three or at least in two databases (Web of Science, PubMed, and Scopus).

The omission of 298 articles after title and abstract review reflects a demanding initial screening process.

Thank you for acknowledging the demanding nature of our initial screening process. We appreciate your understanding of the time required for this task.

As Polanin et al. (2019) highlight, abstract screening is a crucial step in systematic reviews and meta-analyses. It helps narrow down the large number of studies to those that should be fully reviewed, thereby minimizing potential bias and reducing the resource burden in subsequent phases. Efficient and accurate screening is essential, especially when dealing with extensive literature.

We agree that this phase is critical for ensuring the accuracy and reliability of our results and have devoted considerable effort to this process.

Polanin JR, Pigott TD, Espelage DL, Grotpeter JK. Best practice guidelines for abstract screening large‐evidence systematic reviews and meta‐analyses. Res Synth Methods. 2019 Sep;10(3):330–42. doi: 10.1002/jrsm.1354. Epub 2019 Jun 24. PMCID: PMC6771536.

 The study uses various study designs (randomized/non-randomized trials, case series, retrospective studies) providing a broad perspective but this introduces heterogeneity.

Thank you for your insightful comments. We agree that including only prospective randomized clinical trials is ideal for generating strong evidence in a systematic review and mata-analysis.

However, when sufficient studies are not available, including non-randomized trials and retrospective studies can provide valuable information, although this should ideally be the exception rather than the rule.

In our review, due to the limited number of available studies, we included all two-arm studies, encompassing both clinical trials and retrospective studies. We have acknowledged the potential biases associated with this approach and have discussed these limitations in the study.

The review extents a wide temporal range (1952-2023), which could lead to inconsistency due to changes in medical practices over time.

Thank you for your valuable comment. We understand that the extensive temporal range of our review (1952-2023) could introduce inconsistency due to changes in medical practices over time.

Since our review focused on the technique rather than specific time constraints, we included studies that employed the neurotectomy technique in patients, as per our study criteria. Fortunately, most of the studies were laparoscopic, and the surgical technique was consistent across the included studies.

The authors have mentioned high risk of bias in six studies and fair/poor quality in others, however, the high complication rate with neurolysis and neuromodulation permits careful understanding.

Thank you for your thoughtful comments. We acknowledge that our review identified a high risk of bias in six studies and fair or poor quality in others. We agree that the high complication rates associated with neurolysis and neuromodulation necessitate a careful interpretation of the findings.

We have highlighted these issues in the discussion and emphasized the need for cautious interpretation of the results due to the variability in study quality and the potential impact on outcomes.

  Although, neurectomy, neurolysis and neuromodulation shows promise, significant complications and high recurrence rates signifies that it needs further research to establish efficacy and safety.

Thank you for your comments. As correctly noted, we address this issue on page 18, lines 403-409 and page 19 lines 417-419 of our manuscript.

In contrast, the efficacy of neuromodulation in managing endometriosis symptoms appears promising but requires further investigation. With only six studies conducted in recent years, employing various neuromodulation techniques, the evidence is still evolving [9, 26, 40, 45, 46, 56]. However, initial results indicate potential benefits in pain reduction and functional improvement, particularly with sacral neuromodulation [9, 40, 45, 46, 56]. Further research with larger sample sizes and longer follow-up periods is warranted to establish the long-term efficacy and safety profile of neuromodulation in this patient population.

Further research and clinical trials are essential to elucidate the optimal role of neuropelveology interventions in the comprehensive management of endometriosis.

This study needs long-term randomized trials and larger sample sizes addressing the current study limitations with importance on personalized treatment algorithms and predictors of treatment response in patients.  

Thank you for your comment. As noted on page 18, lines 417-419 of our manuscript, we acknowledge that further research and clinical trials are essential to elucidate the optimal role of neuropelveology interventions in the comprehensive management of endometriosis.

 Future studies identifying limitations such as study heterogeneity (study design), variability in surgical techniques (e.g., laparoscopy vs. laparotomy), and limited follow-up data is critical and necessary for contextualizing findings of this study. For instance, a multicenter trial could be designed where all participating centers follow a uniform surgical technique (e.g., laparoscopic neurectomy) and standardized follow-up schedules (e.g., assessments at 6 months, 1 year, and 5 years post-surgery). This approach may help reduce variability and allow for a more accurate comparison of outcomes between the intervention and control groups.

Thank you for your insightful comment. We have revised the limitations section of our study based on your feedback.

Please refer page 19, lines 421-432.

Authors may consider the following papers to be included as references for suggesting future research directions and methodologies.

a.       Design and Implementation by Schulz KF, Altman DG, Moher D et a Lancet; 2001; 357:1191-1194. This paper provides complete strategies on the design and reporting of randomized controlled trials, emphasizing the importance of standardization and rigor in clinical research.

b.      The Importance of Protocol Adherence and Training" by Clavien PA, Barkun J, de Oliveira ML, et al Ann Surg. 2009;250:187– 196. This article discusses the tests and solutions in standardizing surgical techniques in clinical trials, which is crucial for reducing variability and improving the reliability of outcomes.

c.       The Long-term Impact of Surgical Interventions on Endometriosis-Related Pain and Quality of Life: A Systematic Review and Meta-Analysis" by Vercellini P, Viganò P, Somigliana E, et al. This systematic review and meta-analysis focus on the long-term outcomes of surgical interventions for endometriosis, providing valuable insights into the durability of pain relief and quality of life improvements.

Thank you for your valuable suggestion. In the recommendations section of our study, we have incorporated the suggested references.

Please refer page 19, lines 419-423

Unfortunately, the third reference titled "The Long-term Impact of Surgical Interventions on Endometriosis-Related Pain and Quality of Life: A Systematic Review and Meta-Analysis" by Vercellini P, Viganò P, Somigliana E, et al. was not found.